# Meteorological effects on PM$_{2.5}$ change over a receptor region in regional transport of air pollutants: observational study of recent year emission reduction in central China

Xiaoyun Sun[1], Tianliang Zhao[1], Yongqing Bai[2], Shaofei Kong[3], Huang Zheng[3], Weiyang Hu[1], Xiaodan Ma[1], Jie Xiong[2]

[1]Collaborative Innovation Center on Forecast and Evaluation of Meteorological Disasters, Key Laboratory for Aerosol-Cloud-Precipitation of China Meteorological Administration, PREMIC, Nanjing University of Information Science and Technology, Nanjing, 210044, China

[2]Institute of Heavy Rain, China Meteorological Administration, Wuhan, 430205, China

[3]Department of Atmospheric Sciences, School of Environmental Studies, China University of Geosciences (Wuhan), Wuhan, 430074, China

*Correspondence to:* Tianliang Zhao (tlzhao@nuist.edu.cn); Yongqing Bai (baiyq2007@126.com)

**Abstract.** As an important issue in atmospheric environment, the contributions of anthropogenic emissions and meteorological conditions to air pollution have been few assessed over the receptor region in regional transport of air pollutants. In the present study of 5-year observations and modeling, we targeted the Twain-Hu Basin (THB), a large region of heavy PM$_{2.5}$ pollution over central China, to assess the meteorological effects on PM$_{2.5}$ change over a receptor region in regional transport of air pollutants. Based on observations of environment and meteorology over 2015–2019, the Kolmogorov–Zurbenko (KZ) filter was performed to decompose the PM$_{2.5}$ variations into multi-time scale components over the THB, where the short-term, seasonal and long-term components accounted for respectively 47.5 %, 41.4 % and 3.7 % to daily PM$_{2.5}$ changes. The short-term and seasonal components dominated the day-to-day PM$_{2.5}$ variations with long-term component determining the change trend of PM$_{2.5}$ concentrations over recent years. As the emission- and meteorology-related long-term PM$_{2.5}$ components over

the THB were identified, the meteorological contribution to $PM_{2.5}$ declining trend presented the distinct spatial pattern over the THB with northern positive rates up to 61.92 % and southern negative rates down to –24.93 %. The opposite effects of meteorology on $PM_{2.5}$ pollution could accelerate and offset the effects of emission reductions in the northern and southern THB, which is attributed to the upwind diffusion and downward accumulation of air pollutants over the receptor region in regional $PM_{2.5}$ transport. It is noteworthy that the increasing conversion efficiencies of $SO_2$ and $NO_2$ to sulfate and nitrate for secondary $PM_{2.5}$ could offset the effect of $PM_{2.5}$ emission reduction on air pollution in the THB during recent years, revealing the enhancing contribution of gaseous precursor emissions to $PM_{2.5}$ concentrations under controlling anthropogenic emissions of $PM_{2.5}$ and the gaseous precursors over the receptor region in regional transport of air pollutants. Our results highlight the effects of emission mitigation and meteorological changes on source-receptor relationship of region transport of air pollutants with the implication of long-range transport of air pollutants for regional and global environment changes.

## 1.  Introduction

Haze pollution with high levels of $PM_{2.5}$ (fine particulate matters with aerodynamic diameters equal to or less than 2.5 μm) has been a serious problem in atmospheric environment (Peng et al., 2016; Wang et al., 2016) with adverse influences on air quality and human health (Cao et al., 2012; Crouse et al., 2012). In recent years, the large areas over central and eastern China (CEC) have undergone haze pollution with unprecedentedly high $PM_{2.5}$ levels in the regions covering North China Plain (NCP), Yangtze River Delta (YRD), Pearl River Delta (PRD) and Sichuan Basin (SB) (Zhang et al., 2012; Lin et al., 2018a; Guo et al., 2017). In order to improve air quality with reducing air pollutant emissions, Chinese government has implemented an Action Plan of controlling anthropogenic emissions since September 2013 (http://www.gov.cn/xinwen/2018-02/01/content_5262720.htm, last access: August 21, 2021). Surface $PM_{2.5}$ concentrations exhibited 30 %–40 % decreases in CEC over recent years (Xue et al., 2019; Zhang et al., 2019). However, the changes of air pollution are generally co-determined by air pollutant emissions and meteorological conditions. The contributions of changes in meteorology and anthropogenic emissions to the improvement of air quality need to be comprehensively investigated.

PM$_{2.5}$ includes primary particles emitted directly from various sources and secondary particles generated by homogeneous and heterogeneous chemical reactions of gaseous precursor in the atmosphere, depending on the emissions of primary PM$_{2.5}$ and PM$_{2.5}$'s gaseous precursors (Lin et al., 2018b; Du et al., 2020). In addition to the emissions of air pollutants, the meteorological conditions can alter the local accumulation, regional transport, chemical conversion, wet and dry depositions of air pollutants (Lu et al., 2017; Li et al., 2018). Severe haze pollution always occurs in the wintertime under the stagnant meteorological conditions with weak near-surface wind, strong temperature inversion, and high relative humidity in the atmospheric boundary layer, which are favorable for the accumulation of air pollutants to form air pollution (Li et al., 2018; Miao et al., 2015; Tang et al., 2016). Meteorological conditions are closely governed by synoptic circulation by modulating the atmospheric physical and chemical processes including regional transport of air pollutants (Miao et al., 2017; Ning et al., 2019). By regulating synoptic circulation patterns, the climate changes of East Asian monsoons largely influence the seasonal and interannual variations of aerosol concentrations for air pollution over China (Zhu et al., 2012; Jeong and Park, 2017).

Assessments on contributions of anthropogenic emissions and meteorological changes to air quality improvement are an important issue in environmental changes (Pearce et al., 2011; Zhang et al., 2018; Chen et al., 2019). The chemical transport models have been widely used to quantify the meteorological effect on PM$_{2.5}$ variations by a linear additive relationship between sensitivity and base simulations (Mueller and Mallard, 2011; Li et al., 2015b; Zhang et al., 2020). The contribution of meteorological changes to PM$_{2.5}$ decreases was estimated at the averages of 10 %–20 % with the interannual fluctuations of about 5 % in CEC from 2015 to 2019 through a model-based environmental meteorology index (Gong et al., 2021). The accuracy of modeling assessments can be influenced by the uncertainties in emission inventories and the incomplete chemical and physical mechanisms in air pollution simulation (Li et al., 2011). Based on statistical analysis on long-term observational data, it was quantified that the emission control could explain more of the variances in PM$_{2.5}$ than meteorology (Gui et al., 2019), and 12 % of the observed PM$_{2.5}$ decrease was attributed to meteorological drivers in China since 2013 (Zhai et al., 2019). However, the modeling and observational studies have mostly assessed the contribution of emissions and meteorology to regional PM$_{2.5}$ variations in the emission source regions with high anthropogenic emissions of air pollutants, and there have

been few assessments on multi-scale changes of atmospheric environment over the receptor region in regional transport of air

pollutants.

The Twain-Hu Basin (THB), featuring the lower lands (mainly less than 200 m in a. s. l.) of two provinces Hubei and

Hunan in central China (Fig. 1), is surrounded by the high air pollution regions NCP, YRD, PRD and SB. As such, it is a key

receptor region in regional transport of air pollutants from the upstream region driven by East Asian monsoonal winds over

CEC (Shen et al., 2020). Heavy air pollution in the THB with a unique "non-stagnation" atmospheric boundary layer is

aggravated by regional $PM_{2.5}$ transport over CEC (Zhong et al., 2019; Yu et al., 2020). By cohesion with the heavy pollution

region of NCP through distinct transport channels, the regional transport from northern China to central China contributed

70.5 % $PM_{2.5}$ concentrations to a wintertime heavy pollution episode in the THB (Hu et al., 2021). Thus, the contributions of

air pollutant emissions and meteorological conditions to air quality change over this region in central China need to be

specifically assessed with the long-term observations over recent years.

In this study, we investigated the multi-scale changes of $PM_{2.5}$ concentrations over the THB, a key receptor region of

regional $PM_{2.5}$ transport over China from 2015 to 2019 by establishing the statistic model with Kolmogorov–Zurbenko (KZ)

filter, and then evaluated the contributions of anthropogenic emissions and meteorological changes to the declining trends in

$PM_{2.5}$ concentrations over this receptor region in regional $PM_{2.5}$ transport over CEC during the past 5-year emission control.

The analysis of THB's multi-scale air quality changes can improve the understanding of the effects of emission mitigation and

meteorological changes on environmental change with regional transport of air pollutants.

## 2.   Data and methods

### 2.1  Data

In order to analyze air quality changes in the THB, the observational data of hourly $NO_2$, $SO_2$ and $PM_{2.5}$ concentrations

from 2015 to 2019 were collected from the national air quality monitoring network (http://www.mee.gov.cn/, last access:

August 21, 2021). The air quality observation data are under quality control, based on China's national standard of air quality

observation.

The data of meteorological observations in the THB were sourced from the weather monitoring network of China Meteorological Administration (http://data.cma.cn/, last access: August 21, 2021), including air temperature (T), relative humidity (RH), sea level pressure (SLP), wind speed (WS) and precipitation (Pre) with temporal resolutions of 3 h.

## 2.2 KZ filter

To better understand the multi-time scale variations of PM$_{2.5}$ and the relations to air pollutant emissions and meteorological drivers, KZ filter (Rao and Zurbenko, 1994; Seo et al., 2018) is used to separate the daily data into multi-scale components, based on an iterative moving average that removes high frequency variations in the data with the applications in study of air pollutants, especially O$_3$ and PM$_{2.5}$ variations (Chen et al., 2019; Ma et al., 2016; Seo et al., 2014; Zheng et al., 2020).

The KZ filter $KZ_{m,p}$ with the length of moving average window *m* and the number of iterations *p*, can remove the high-frequency component of period smaller than the effective filter width N ($\geq$ m $\times$ $p^{1/2}$). The KZ filter is applicable to the time series with missing data owing to the iterative moving average process, which provides a high accuracy level to compare with the wavelet transform method (Eskridge et al., 1997). By comparing different sets of moving average *m* and number of iterations *p*, it was found that the decomposed time series using $KZ_{15,5}$ filter exhibited no white noise (short-term component), and the trend of long-term component derived with $KZ_{365,3}$ filter corresponded approximately to the interannual trend of the original data (Rao and Zurbenko, 1994; Eskridge et al., 1997). Based on the spectral decompositions of the daily observational data and three components, the power spectra of daily observational data in periods less than 33 days and longer than 632 days (1.7 years) have been well reproduced by short-term and long-term components, and seasonal component represents well the seasonal variations, i.e., periods between 33 days and 1.7 years (Seo et al., 2018). Thus we applied $KZ_{15,5}$ and $KZ_{365,3}$ filters to remove the variations with the periods shorter than 33 days and 1.7 years in this study.

A meteorological or environmental variable $X(t)$ observed in time series *t* can be decomposed into the short-term component $X_{ST}(t)$ and the baseline component $X_{BL}(t)$ presenting as:

$$X(t) = X_{ST}(t) + X_{BL}(t). \tag{1}$$

The baseline component $X_{BL}(t)$ is obtained by applying the $KZ_{(15,5)}$ filter to $X(t)$, removing the short-term component $X_{ST}(t)$ with the temporal period shorter than 33 days from the observed data, expressing with:

$$X_{BL}(t) = KZ_{(15,5)}[X(t)]. \tag{2}$$

The baseline component $X_{BL}(t)$ also can be separated into the daily climatic averages $X_{BL}^{clm}$ over the study period occupying most of the seasonality in $X_{BL}(t)$ and the residual $\varepsilon(t)$:

$$\varepsilon(t) = X_{BL}(t) - X_{BL}^{clm}. \tag{3}$$

To obtain the long-term component $X_{LT}(t)$ by removing the variations with the temporal period shorter than 1.7 years, the $KZ_{365,3}$ filter is applied to $\varepsilon(t)$ expressing as follows:

$$X_{LT}(t) = KZ_{(365,3)}[\varepsilon(t)] \tag{4}$$

with the short-term component

$$X_{ST}(t) = X(t) - X_{BL}(t) \tag{5}$$

and the seasonal component

$$X_{SN}(t) = X_{BL}(t) - X_{LT}(t). \tag{6}$$

The KZ filter was used to separate the daily surface $PM_{2.5}$, $NO_2$ and $SO_2$ concentrations into short-term, seasonal and long-term components in this study. The short-term component presents a synoptic-scale variation of meteorological influences, which could control local accumulation and regional transport of air pollutants (Seo et al., 2017), partly associated with short-term fluctuations in air pollutant emissions (Russell et al., 2010). The seasonal and long-term components are attributable to the variations in air pollutant emissions related to human activities as well as the seasonal and interannual changes in meteorological conditions (Kim et al., 2018).

## 2.3 Multiple linear regression of air pollutant changes with meteorological variables

By altering the local accumulation, regional transport, chemical conversion, wet and dry depositions of air pollutants, the

meteorological factors such as wind, RH, T, air pressure and Pre could exert significant impacts on $PM_{2.5}$ changes (Sun et al., 2013; Li et al., 2018; Chen et al., 2020b). Therefore, with the multiple factors of the baseline components of 10-m WS, 2-m RH, 2-m T, SLP and Pre calculated by Eq. (2), a multiple linear regression equation was stepwise established for the baseline component of $PM_{2.5}$ as follows:

$$PM_{2.5BL_{MLR}}(t) = a_0 + \sum_i a_i \, \text{MET}_{BL_i}(t), \tag{7}$$

where $\text{MET}_{BL_i}(t)$ ($i \epsilon [1,5]$) is the baseline component of the meteorological variable $i$ with $i=1,2,3,4,5$ respectively for $WS_{BL}(t)$, $RH_{BL}(t)$, $T_{BL}(t)$, $SLP_{BL}(t)$, $Pre_{BL}(t)$. We fit the regression coefficient $a_i$ for each meteorological variable and the intercept $a_0$. The residual $\varepsilon_{PM_{2.5}}$ between $PM_{2.5BL}$ and $PM_{2.5BL_{MLR}}$ regressed with the multiple linear equation (7) is given as:

$$\varepsilon_{PM_{2.5}}(t) = PM_{2.5BL}(t) - PM_{2.5BL_{MLR}}(t). \tag{8}$$

$\varepsilon_{PM_{2.5}}$ contains not only the variability of $PM_{2.5}$ related to long-term changes in air pollutant emissions but also the minor seasonal change of $PM_{2.5}$ attributable to unconsidered meteorological influences in the multiple linear regression. By removing the minor seasonal change from $\varepsilon_{PM_{2.5}}$ with the $KZ_{365,3}$ filter, the emission-related long-term component $PM_{2.5LT}^{emiss}(t)$ can be isolated as follows:

$$PM_{2.5LT}^{emiss}(t) = KZ_{(365,3)}\big[\varepsilon_{PM_{2.5}}(t)\big]. \tag{9}$$

Here the long-term component of surface $PM_{2.5}$ concentrations can be further separated into the emission- and meteorology-related long-term components with Eqs. (9) and (4) (Seo et al., 2018). Similarly, the multi-time scale variations in $SO_2$ and $NO_2$ with long-term variations related to changes in air pollutant emissions and meteorological drivers are decomposed by KZ filter with multiple linear regression. Seo et al. (Seo et al., 2018) described the details of this method.

## 3. Results and discussion

### 3.1 Verification of PM$_{2.5}$ decompositions in multi-scale variations

The daily $PM_{2.5}$ concentrations observed in 14 sites over the THB (Fig. 1) were decomposed into short-term, seasonal and

long-term components with Eqs. (4), (5) and (6) of the KZ filter. To verify the decomposition results, the spatial distribution of total contributions of short-term, seasonal and long-term $PM_{2.5}$ components to the total variances of observed daily changes in $PM_{2.5}$ concentrations over 2015–2019 were shown in Figure 2a. The larger the total variance, the more independent the three components are of each other (Chen et al., 2019). The sum of the long-term, seasonal and short-term components contributed 91.4 %–94.4 % to the total variance with the regional averages of 92.7 % (Fig. 2), reflecting a satisfactory verification of the KZ filtering results.

Based on the $PM_{2.5}$ decomposition results of KZ filter, the short-term, seasonal and long-term components respectively accounted for 34.8 %–53.8 %, 29.2 %–56.3 % and 0.2 %–9.8 % of the total variances of daily $PM_{2.5}$ changes in the THB over recent years (Figs. 2b, 2c and 2d), reflecting the different patterns of multi-time scale variations of $PM_{2.5}$ over this region in central China with diverse effects of emissions and meteorology. The regional contributions of short-term, seasonal and long-term components were averaged respectively with 47.5 %, 41.4 % and 3.7 % to daily $PM_{2.5}$ changes over the THB (Fig. 2), which could be reasonably verified that the daily variation in atmospheric pollutant was generally dominated by short-term and seasonal components with long-term component determining the change trend (Ma et al., 2016; Yin et al., 2019a).

The short-term, seasonal and long-term $PM_{2.5}$ components were averaged in 14 sites of the THB to characterize the temporal variations of three components in the THB for 2015–2019 (Fig. 3). The correlation coefficients of 0.05, 0.01 and 0.04 among the decomposed short-term, seasonal and long-term components were near zero, indicating the orthogonal decomposition of multi-time scale components (Eskridge et al., 1997). According to the decomposed long-term, seasonal and short-term components demonstrated in Fig. 3, the notable peaks of decomposed seasonal and short-term components were highly consistent with the peaks of $PM_{2.5}$ concentrations in the original observed data, which further proved a reasonable decomposition of the multi-scale components of $PM_{2.5}$ change over 2015–2019.

The observed daily $PM_{2.5}$ exhibited a distinct daily variation, with an overlapping of high frequency variations, which could be caused by mesoscale and synoptic scale meteorological processes (Ma et al., 2016). The short-term component of $PM_{2.5}$ fluctuated frequently with a significantly positive correlation to the daily change of $PM_{2.5}$ (r = 0.68, p<0.05), indicating

an important role of the short-term component with the temporal period < 33 days in the day-to-day variations of $PM_{2.5}$ concentrations in the THB (Fig. 3a).

The notable peaks of $PM_{2.5}$ seasonal component emerged in winters were highly in keeping with the peaks of observed daily $PM_{2.5}$ concentrations (Fig. 3b). A close linkage with the significant correlation coefficient of 0.75 (p<0.05) was found between the changes of $PM_{2.5}$ seasonal component and daily $PM_{2.5}$ concentrations, which could reflect a significant modulation of the $PM_{2.5}$ seasonal oscillations to the day-to-day variations of $PM_{2.5}$, driven by the seasonal shift of East Asian summer and winter monsoons as well as the seasonal change of anthropogenic emissions (Zhu et al., 2012; Jeong and Park, 2017). The change of long-term component of $PM_{2.5}$ exhibited a steadily declining trend over 2015–2019 (Fig. 3c), which was consistent with the interannual trend of observed $PM_{2.5}$ concentrations under the sustained impact of emission control (Zhang et al., 2019; Xu et al., 2020). The correlation coefficient (r = 0.24, p<0.05) of long-term $PM_{2.5}$ component with the observed daily $PM_{2.5}$ change was much smaller than those of short-term and seasonal $PM_{2.5}$ components, implying less influence of emission reduction on the daily $PM_{2.5}$ change and air pollution frequency, although the declining trend in $PM_{2.5}$ was determined by anthropogenic emission reduction.

In previous studies, chemical transport models and statistical methods were both used to assess the changes in air pollution attributable to emissions and meteorology (Xiao et al., 2021). Significant declines in emission-related $PM_{2.5}$ concentrations occurred in central China (Wang et al., 2019; Chen et al., 2020a), and the meteorology offset the impact of emission reduction in typical years of unfavorable meteorological conditions (Xu et al., 2020; Gong et al., 2021). The regional averaged emission- and meteorology-related long-term components as well as the long-term component over the THB are displayed in Fig. S1a, implying the steadily declining trend of $PM_{2.5}$ and the dominant impact of emission reduction on long-term $PM_{2.5}$ changes, which is consistent with the previous studies using multiple linear regression model for central China (Fig. S1b). The meteorology-related long-term component is positive value in certain periods, implying the significant modulating effect of meteorology on $PM_{2.5}$ decline in the THB.

## 3.2 Multiple linear regressions of $PM_{2.5}$, $SO_2$ and $NO_2$ with meteorological drivers

Since the short-term variations in meteorological variables were excluded, the correlations between baseline components of $PM_{2.5}$ and meteorological variables were only related to their seasonal and long-term components, affected by regional climate of East Asian monsoons rather than synoptic-scale meteorological processes. Based on our understanding of chemical and physical processes of diffusive transport, chemical transformation, emissions and depositions of $PM_{2.5}$ in the atmosphere, the dominant meteorological factors for changing $PM_{2.5}$ concentrations over china are wind speed, relative humidity, air temperature, atmospheric pressure and precipitation (Chen et al., 2020b). We examined the significant correlations between baseline components of air pollutant concentrations and selected a set of meteorological factors, including air temperature, wind speed, precipitation, relative humidity, and air pressure (Tables S1-S3). The meteorological parameters selected in this study are consistent with the previous studies (Chen et al., 2020b).

Generally, the baseline components of air pollutants were negatively correlated with baseline components of wind speed ($WS_{BL}$) and positively correlated with baseline components of sea level pressure ($SLP_{BL}$) (Tables S1–S3), which could be attributed to the ventilation effect of wind and stagnant condition of meteorology in high-pressure systems, restraining the horizontal and vertical dispersions of air pollutants (Hsu and Cheng, 2016; Wang et al., 2016; Miao et al., 2017). Although wind speed exerts a negative influence of on $PM_{2.5}$ concentrations over the emission source region, increasing wind speed might cause the accumulation of $PM_{2.5}$ concentrations over the downwind region of emission sources (Chen et al., 2020b), which led to the inconsistent influence of $WS_{BL}$ in the region of central China (Tables S1–S3). Under surface high air temperature conditions, there are strong thermal activities such as turbulence, making an accelerated dispersion of air pollutants (Yang et al., 2016b). The negative influence of $RH_{BL}$ and $T_{BL}$ on $PM_{2.5BL}$, $SO_{2BL}$ and $NO_{2BL}$ mainly reflected the effect of seasonal cycle in East Asian winter and summer monsoons, whereas the influence of precipitation on air pollutants was more straightforward than other meteorological parameters, negatively influencing surface pollutant concentrations through the precipitation washout of air pollutants (Tables S1–S3).

To isolate emission-related long-term components from long-term components of $PM_{2.5}$, $NO_2$ and $SO_2$, the stepwise

multiple linear regressions of $PM_{2.5BL}$, $SO_{2BL}$ and $NO_{2BL}$ respectively with baseline components of meteorological parameters

($T_{BL}$, $WS_{BL}$, $RH_{BL}$, $SLP_{BL}$ and $Pre_{BL}$) were conducted with Eq. (7) in 14 sites, by adding and deleting meteorological variables

based on the independent statistical significance to obtain the best model fit (Draper, 1998). We evaluated the $PM_{2.5BL}$, $SO_{2BL}$

and $NO_{2BL}$ fitted by the multiple linear regression models with KZ decomposition (Table 1). The multiple linear regressions

explained $PM_{2.5BL}$, $SO_{2BL}$ and $NO_{2BL}$ with adjusted determination coefficients (Adj. $R^2$) of 0.5695–0.8093, 0.0630–0.4592 and

0.6304–0.8669 passing the confidence level of 99 % in all the THB sites, confirming the reasonable construct of multiple linear

regressions. The Adj. $R^2$ of multiple linear regression for $SO_{2BL}$ were lower than those of $PM_{2.5BL}$ and $NO_{2BL}$, which might be

attributed to the larger impact of $SO_2$ emission control on the seasonal and long-term $SO_2$ variations. In general, the variations

of meteorological drivers can well reproduce the meteorology-related seasonal and long-term variations of $PM_{2.5}$, $SO_2$ and

$NO_2$ in the THB (Table 1).

### 3.3  Interannual variations in air pollutants observed over the THB

$PM_{2.5}$ consist of chemical components generated in the complex physical and chemical processes (Li et al., 2015a).

Primary particles are emitted directly from anthropogenic (e.g., industry, power plants, and vehicles) and natural (e.g., outdoor

biomass burning and dust storms) sources. Secondary particles (e.g. sulfate and nitrate) are converted with chemical reactions

of the precursor gases (e.g., $SO_2$ and NOx), which are mainly produced by human activities (Li et al., 2015a; Yang et al.,

2016a). Therefore, in addition to the reductions in primary particulate emissions, control of the secondary aerosol precursor

emissions is of great importance in mitigating air pollution.

The interannual variations of the ratios in annual mean $PM_{2.5}$, $SO_2$ and $NO_2$ concentrations relative to the annual averages

in 2015 over the THB are displayed in Figure 4. The declines of $PM_{2.5}$ and $SO_2$ in 2019 averaged over the THB were –26 %

and –68 % relative to 2015, while the decrease ratio in $NO_2$ was only –8 % over this region. The observed $SO_2$ concentrations

had a steeper decrease than $PM_{2.5}$ and $NO_2$, possibly because the dominant source sectors (i.e., power and industry) of $SO_2$

significantly reduced their emissions (Zheng et al., 2018). The power sector was the major contributor to emission reduction

but only accounted for one-third of $NO_X$ emissions and the contribution of transportation to $NO_X$ emissions was estimated to have increased over recent years (Zheng et al., 2018). The interannual variations in emissions for China were calculated from MEIC (Zheng et al., 2018), as well as the annual total emissions of $SO_2$ and $NO_x$, PM in the THB region reported by National Bureau of Statistic of China (http://www.stats.gov.cn/tjsj/ndsj/, last access: January 17, 2022), presenting the rapid decline of $SO_2$ emissions in the THB than changes of $PM_{2.5}$ and $NO_x$ emissions (Fig. S2). The declining trend of anthropogenic emissions estimated from emission inventories can support the explanation of the changes in air pollutant concentrations.

Figure 5 shows the spatial distributions of 5-year averaged concentrations, the linear trends and the change rates in interannual variations of $PM_{2.5}$, $SO_2$ and $NO_2$ observed in the THB over 2015–2019. The change rates (% $yr^{-1}$) were calculated with the linear trends by dividing with temporal-mean concentrations of air pollutants at the observation sites for the analysis period in Figure 5. The 5-year averaged $PM_{2.5}$ concentrations over the THB exceeded the Chinese National secondary air quality standard of 35 μg $m^{-3}$ for annual mean $PM_{2.5}$ concentrations (Fig. 5a), while $SO_2$ and $NO_2$ concentrations reached the secondary standards of 60 μg $m^{-3}$ and 40 μg $m^{-3}$ in annual mean $SO_2$ and $NO_2$ concentrations at most sites over the THB (Figs. 5d and 5g). Specifically, the 5-year averaged $NO_2$ concentrations exceeded 40 μg $m^{-3}$ in WH (Wuhan), the mega-city in central China, that might be attributable to the large amounts of traffic transportation. From 2015 to 2019, both $PM_{2.5}$ and $SO_2$ decreased at all sites over the THB (Figs. 5b and 5e), whereas $NO_2$ trends were changed from mostly negative to positive in some sites (Fig. 5h), possibly due to the spatial disparity of NOx emissions in traffic sectors (Zheng et al., 2018). The comparison among the change rates of $PM_{2.5}$, $SO_2$ and $NO_2$ in the THB presented the largest decreases of $SO_2$ with –20% – –40% $yr^{-1}$ over the five years (Figs. 5c, 5f and 5i), reflecting the effective control of $SO_2$ emissions in terms of primary gaseous pollutants.

There were obvious decreases in regional mean $PM_{2.5}$, $SO_2$ and $NO_2$ concentrations over the THB (Fig. 4), while the declining degree of $PM_{2.5}$ and $SO_2$ varied from site to site over the THB and the change trends in $NO_2$ were weak negative and even positive in certain sites (Figs. 5c, 5f and 5i). These interannual changes of air pollutants in the THB over recent years were investigated with the emission- and meteorology-related long-term components of air pollutants in the next sections.

**3.4 Effects of NO₂ and SO₂ emission reductions on PM₂.₅ change trends**

The declining trend of $PM_{2.5}$ in China could be partly attributed to the reduced $NO_x$ and $SO_2$ concentrations for producing the secondary aerosols (Zhang et al., 2018). The reduction rates of anthropogenic emissions have markedly accelerated after 2013, decreasing by 59% for $SO_2$, 21% for NOx and 33% for $PM_{2.5}$ during 2013–2017 over China (Zheng et al., 2018). In order to assess the effect of changing precursor pollutant emissions on $PM_{2.5}$ declines, we compared the linear trends of emission-related long-term components of $PM_{2.5}$, $NO_2$ and $SO_2$ decomposed based on Eq. (9) over the THB for 2015–2019 (Fig. 6). The distinct declining trends of emission-related long-term $PM_{2.5}$ and $SO_2$ components as well as the variable trends of emission-related long-term $NO_2$ components were distributed basically consistent with the positive and negative trends in the interannual variations of air pollutant concentrations in the THB (Fig.5 (middle column); Fig. 6), demonstrating that the local emissions of air pollutants could spatially dominate the long-term variations of air pollutants in central China, especially the increasing trends in $NO_2$ at some THB sites.

$PM_{2.5}$ concentrations are changed by emissions of both primary $PM_{2.5}$ and $PM_{2.5}$'s gaseous precursors. As major gaseous precursors, $SO_2$ and $NO_2$ can be oxidized to convert nitrate and sulfate for secondary $PM_{2.5}$ (Li et al., 2015a). To investigate the effects of emission reductions on the interannual variations of $PM_{2.5}$, $NO_2$ and $SO_2$ over recent years, the ratios of change trends in long-term ($k_{LT}$) and emission-related long-term ($k_{emiss}$) components of $PM_{2.5}$, $SO_2$ and $NO_2$, in the THB over 2015–2019 were demonstrated in Figure 7, where the long-term and emission-related long-term components of $PM_{2.5}$, $SO_2$ and $NO_2$ were calculated with Eqs. (4) and (9). The trend ratios $k_{LT}/k_{emiss}$ <1 indicated the more obvious downward trend of emission-related long-term variations than the long-term trend of air pollutant concentrations, which might be attributed to the offsetting effect of meteorological conditions on emission reduction in air quality change, whereas the long-term trend of air pollutant concentrations was more significant than the emission-related long-term trend with $k_{LT}/k_{emiss}$ >1, reflecting the synchronous impacts of anthropogenic emissions and meteorology on the long-term trend in air pollutant change. In addition, the trend ratios $k_{LT}/k_{emiss}$ >1 and $k_{LT}/k_{emiss}$ <1 of $PM_{2.5}$'s gaseous precursors $SO_2$ and $NO_2$ could reflect the high and weak

efficiencies of $SO_2$ and $NO_2$ converting to sulfate and nitrate in the production of secondary $PM_{2.5}$ during air pollutant emission reduction. The notable differences in Figure 7 were spatially distributed with the trend ratios $k_{LT}/k_{emiss} > 1$ and $k_{LT}/k_{emiss} < 1$ in $PM_{2.5}$, $SO_2$ and $NO_2$ concentrations under the same meteorological conditions, indicating the different influences of emissions on the long-term variations of $PM_{2.5}$, $SO_2$ and $NO_2$ in the THB during recent years. The reduction in $PM_{2.5}$ emissions was a primary cause for the long-term declines in $PM_{2.5}$ concentrations in the THB, even though the meteorological changes might offset the effects of emission reduction on air quality improvement over the southern THB (Figs. 6 and 7). It is noteworthy that the trend ratios $k_{LT}/k_{emiss} < 1$ of $PM_{2.5}$ were accompanied with $k_{LT}/k_{emiss} > 1$ of $SO_2$ and $NO_2$ at the downwind southern THB sites with both negative $k_{LT}$ and $k_{emiss}$ (Fig. 7, Table S4), which could imply the increasing conversion efficiency of $SO_2$ and $NO_2$ to sulfate and nitrate for secondary $PM_{2.5}$ during the reductions of air pollutant emissions over recent years. In the upwind northern THB sites, the $k_{LT}/k_{emiss} > 1$ of $PM_{2.5}$ were accompanied with $k_{LT}/k_{emiss} > 1$ of $SO_2$ and $NO_2$ with obviously facilitating effect of meteorology on $PM_{2.5}$ decline (Fig. 7, Table S4), revealing the underlying effect of regional transport of air pollutants on the spatial distribution of conversion efficiency of gaseous precursor to secondary $PM_{2.5}$.

In order to further assess the effect of gaseous precursor emissions on $PM_{2.5}$ declines during recent 5-year air pollution mitigation, we selected 7 and 9 sites in the THB with the decreasing trends of emission-related long-term $SO_2$ and $NO_2$ components below –0.5 and 0.0 μg m$^{-3}$ 100d$^{-1}$ respectively (Table S4) to compare the trend ratios $k_{LT}/k_{emiss}$ of $PM_{2.5}$, $NO_2$ and $SO_2$ for 2015–2019 (Fig. 8). The significantly negative linear correlations between changes in $k_{LT}/k_{emiss}$ of gaseous precursors ($SO_2$ and $NO_2$) and $PM_{2.5}$ could present the connection of $k_{LT}/k_{emiss} > 1$ for $NO_2$ and $SO_2$ with $k_{LT}/k_{emiss} < 1$ for $PM_{2.5}$, which confirmed the fact that the high conversion efficiency of $SO_2$ and $NO_2$ to sulfate and nitrate could offset the role of $PM_{2.5}$ emission reduction in controlling $PM_{2.5}$ pollution. The decreasing emissions of gaseous precursors drove faster oxidation of $NO_2$ and $SO_2$ to nitrate and sulfate components of $PM_{2.5}$ in the source regions of air pollution in China (Zhai et al., 2021; Huang et al., 2021). This study identified the enhancing contribution of gaseous precursors to $PM_{2.5}$ concentrations with reducing anthropogenic emissions of air pollutants over the receptor region in regional $PM_{2.5}$ transport.

There are a few sources of uncertainty in the discussion of chemical transformation, for example in the separation of emission- and meteorology-related long-term components and in the selection of observational sites. Our results point to better understand the offsetting effect of $SO_2$ and $NO_2$ oxidized to secondary particles on $PM_{2.5}$ mitigation during the emission reduction in the THB. The long-term changes in $PM_{2.5}$ are also caused by the emission variations of primary components like black and organic carbon, in addition to the chemical transformation of gaseous precursors. The difference in the emission of different primary pollutants may also lead to modifications in $k_{LT}/k_{emiss}$ of $PM_{2.5}$. However, due to the current lack of long-term observation of $PM_{2.5}$ components in the THB, the influence of emission variations of primary components on long-term changes in $PM_{2.5}$ concentrations is not assessed in our study. Further work with long-term observational data of $PM_{2.5}$ components like black and organic carbon could be conducted to quantify the influence of emissions of primary components and chemical transformation of gaseous precursors on $PM_{2.5}$ changes.

### 3.5 Meteorological contribution to PM₂.₅ change trends

As the air pollutant change trend is assumed to generally consist of emission- and meteorology-related changes (Seo et al., 2018; Yin et al., 2019b), the meteorological contribution rate $Con_{met}$ to long-term $PM_{2.5}$ change trend is calculated with the following equation:

$$Con_{met} = \frac{k_{LT} - k_{emiss}}{k_{LT}} \times 100\%. \tag{10}$$

Here, $Con_{met}$ (in %) is estimated with the linear trends $k_{LT}$ of long-term component $PM_{2.5LT}(t)$ and $k_{emiss}$ of emission-related long-term component $PM_{2.5LT}^{emiss}(t)$. $PM_{2.5LT}(t)$ and $PM_{2.5LT}^{emiss}(t)$ are respectively calculated with Eqs. (4) and (9).

To quantitatively assess the meteorological contribution to the $PM_{2.5}$ declining trends, the linear trends $k_{LT}$ and $k_{emiss}$ with the meteorological contribution rate $Con_{met}$ in Eq. (10) were presented in Table S5 for 14 sites over the THB during 2015-2019. All the trends $k_{LT}$ and $k_{emiss}$ respectively in $PM_{2.5LT}(t)$ and $PM_{2.5LT}^{emiss}(t)$ were negative over the THB (Table S5), indicating the significant effect of emission reductions on $PM_{2.5}$ declining trends for improving regional air quality in central China. By comparing the $PM_{2.5}$ declining trends $k_{emiss}$ and $k_{LT}$ from site to site (Table S5), the positive and

negative contributions of meteorological variations to PM$_{2.5}$ change trends over recent years were determined with the positive and negative differences between $k_{emiss}$ and $k_{LT}$ with the distinct meteorological influences on the change of THB's regional environment.

The spatial distribution of meteorological contribution rates $Con_{met}$ to long-term PM$_{2.5}$ declining trend presented the unique pattern of northern positive and southern negative values over the THB (Fig. 9), with high positive contributions in northern sites XY (61.92 %) and EZ (37.31 %) as well as low negative contributions in southern sites CD (–24.93 %) and CS (–23.03 %). It is worth mentioning that the contribution rates of meteorological variations show great spatial disparities at a small scale, i.e., EZ, HG and HS, which seems not be induced by the variation in synoptic weather or meteorological conditions. The underlying surface conditions dominate the near-surface meteorological conditions in the atmospheric boundary layer at a small scale (Wang et al., 2017). The topography and land use of HG, HS, EZ and surrounding regions vary distinctly with underlying surface conditions of plain, lakes and hilly area. The underlying surface of observational sites with different near-surface meteorology effectively influence the local accumulation, chemical transformation, dry and wet depositions of air pollutants (Bai et al., 2022). Therefore, the heterogeneity of meteorological contribution to PM$_{2.5}$ at such a small spatial scale might be attributed to the local meteorological conditions in the atmospheric boundary layer, which is largely affected by the underlying surface changes.

Comparing with the statistical studies using synthetic data of meteorological influence on regional PM$_{2.5}$ changes in central China with the meteorological contribution from –45.5 % to 29.0 % over recent years (Gong et al., 2021; Chen et al., 2020a), the PM$_{2.5}$ pollution over the THB was affected contrarily by meteorological drivers with the northern positive and southern negative contribution from 2015 to 2019 (Fig. 9). The meteorological change could accelerate and offset the effects of emission reductions on PM$_{2.5}$ declining trends in the northern and southern THB, which might be attributed to regional transport of air pollutants conducive to the upwind diffusion and downward accumulation of air pollutants respectively over the northern and southern THB under the declining wind of East Asian monsoons over recent years (Hu et al., 2020; Zhong et al., 2019).

**3.6 Meteorological contribution to PM$_{2.5}$ changes validated with WRF-Chem modeling**

The above observational study investigated the meteorological influence on the changes in PM$_{2.5}$ concentrations in the THB using KZ filter, with concluding the large impact of meteorology on the PM$_{2.5}$ changes over 2015–2019. To validate this conclusion of analyses with KZ filter, we designed three sets of modeling experiments CTRL, SENS-MET and SENS-EMI (Table S6) for December of 2015–2019, respectively driven with the changing meteorology and anthropogenic emissions over 2015–2019, the fixed meteorological conditions and anthropogenic emissions of 2015 with atmospheric chemical model WRF-Chem (Weather Research and Forecasting model with Chemistry). Air pollutant emission inventories, modeling configuration, experiment design and modeling verification were described in the supplement. The modeling verification of experiments CTRL indicated that PM$_{2.5}$ and meteorology were reasonably reproduced by the WRF-Chem simulation (Figs.S4–S5, Table S7), and the designed three sets of modeling experiments CTRL, SENS-MET and SENS-EMI could be used in the further analyses of emission and meteorological impact on PM$_{2.5}$ change over 2015–2019 to confirm the results of KZ filter.

We derived the effect of meteorology by comparing the simulated PM$_{2.5}$ concentrations in three sets of experiments CTRL, SENS-MET and SENS-EMI (Table S6). The relative contribution of meteorology to the interannual changes of PM$_{2.5}$ concentrations was calculated with a linear additive relationship of contributions of meteorology and emission in the following equations:

$$Con_{MET} = \frac{k_{MET}}{k_{CTRL}} \tag{11}$$

$$Con_{EMI} = \frac{k_{EMI}}{k_{CTRL}} \tag{12}$$

$$RCon_{MET} = \frac{Con_{MET}}{Con_{MET} + Con_{EMI}} \times 100\% \tag{13}$$

$k_{CTRL}$, $k_{MET}$ and $k_{EMI}$ represent the trends in interannual changes of PM$_{2.5}$ concentrations simulated by the experiments CTRL, SENS-MET and SENS-EMI, respectively. $Con_{MET}$ and $Con_{EMI}$ are the contribution of meteorology and emission, and $RCon_{MET}$ is the contribution rate (%) of meteorology to interannual changes of PM$_{2.5}$ concentrations (Zhang et al., 2020).

Based on WRF-Chem modeling experiments, we assessed the impact of meteorological changes on interannual PM$_{2.5}$ variations from 2015 to 2019 with *Eqs. (11–13)*. The relative contribution of meteorology to interannual PM$_{2.5}$ variations

displayed the regional pattern of northern positive and southern negative values over the THB (Fig. 10), confirming the impact of meteorological changes by accelerating and offsetting the effects of emission reductions on $PM_{2.5}$ declining trends in the northern and southern THB, respectively. The general spatial distribution of meteorological contribution rates to $PM_{2.5}$ declining trends from the WRF-Chem simulation was consistent with the results using KZ filter (Figs. 9 and 10), validating the results with KZ filter that meteorological drivers exerted a contrary impact of northern positive and southern negative contribution on long-term changes of $PM_{2.5}$ concentrations in the THB.

## 4. Conclusions

The meteorological effect on multi-scale changes of atmospheric environment has been few assessed for the receptor region in regional transport of air pollutants. In this study of observations and modeling, we targeted the THB, a large region of heavy $PM_{2.5}$ pollution over central China, to assess the meteorological effects on $PM_{2.5}$ changes over a receptor region in regional transport of air pollutants during recent five years. The study results provide insights in the effects of emission mitigation and meteorological changes on source-receptor relationship of long-range transport of air pollutants for regional and global environment changes.

This study decomposed the observed $PM_{2.5}$ concentrations into multi-time scale components with a modified KZ filter, to better understand the $PM_{2.5}$ variations with the short-term, seasonal and long-term components accounting for respectively 47.5 %, 41.4 % and 3.7 % to observed $PM_{2.5}$ changes. The short-term and seasonal $PM_{2.5}$ components dominated the daily $PM_{2.5}$ changes and long-term component determined the trend of $PM_{2.5}$ change over recent years. The long-term components of $PM_{2.5}$, $SO_2$ and $NO_2$ were further isolated into emission- and meteorology-related long-term components with multiple linear regressions, to figure out the contributions of emission and meteorology to $PM_{2.5}$ decline in the THB over 2015–2019. The reduction in anthropogenic emissions was the primary cause for long-term decline in $PM_{2.5}$ concentrations and the meteorological changes moderated the $PM_{2.5}$ variations in the THB. As the receptor region of regional $PM_{2.5}$ transport, the impact of diverse meteorological conditions on long-term trend of $PM_{2.5}$ changes displayed unique regional pattern of northern

positive rates up to 61.92 % and southern negative rates down to –24.93 %. The change of meteorological conditions could accelerate and offset the effects of emission reductions on $PM_{2.5}$ declining trends in the northern and southern THB, which could be attributed to the upwind diffusing and downward accumulating roles of regional transport pathway on air pollutants in the THB. In terms of gaseous precursor emissions, the increasing conversion efficiency of $SO_2$ and $NO_2$ to sulfate and nitrate for secondary $PM_{2.5}$ could offset the role of emission reduction in controlling air pollution, and the contribution of gaseous precursors to secondary $PM_{2.5}$ enhanced with the reducing anthropogenic emissions of air pollutants over this receptor region.

This study exposed the impact of anthropogenic emissions and meteorological conditions on the $PM_{2.5}$ decline over a receptor region in regional transport of air pollutants in central China. The effect of regional transport on $PM_{2.5}$ pollution over the receptor region is found differing from that over the source regions with high anthropogenic emissions. We took considered the uncertainties induced by statistical methods as systematic biases and explained the offsetting effect of enhancing oxidation of gaseous precursors to secondary particles on $PM_{2.5}$ decline during the stringent emission controls. The changes in data coverage and the meteorological parameter selection would largely influence the quantitative estimation of contributions of meteorology and emissions. Further work could be desired with climate analyses of long-term data of fine meteorological and environmental observations and more comprehensively modeling of chemical and physical processes in the atmosphere to generalize the assessment on the effects of emission mitigation and meteorological changes on source-receptor relationship of region transport of air pollutants.

*Data availability.* Data used in this paper can be provided upon request from Xiaoyun Sun (sunxy6362@126.com) or Tianliang Zhao (tlzhao@nuist.edu.cn).

*Author contributions.* TZ and XS conceived the study. YB provided the observation data and analysis. XS designed the graphics and wrote the manuscript with help from TZ, YB and SK. HZ, WH, XM and JX were involved in the scientific discussion. All authors commented on the paper.

*Competing interests.* The authors declare that they have no conflict of interest.

*Acknowledgement.* This research was financially funded by grants from National Natural Science Foundation of China (41830965; 42075186; 91744209) and the National Key R & D Program Pilot Projects of China (2016YFC0203304).

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

**Table 1** Adjusted determination coefficients (Adj. $R^2$) between the baseline components decomposed by KZ filter and fitted

with multiple linear regressions respectively for $PM_{2.5BL}$, $SO_{2BL}$ and $NO_{2BL}$ in 14 sites over the THB. All Adj. $R^2$ passing the

confidence level of 99%.

| Sites | Adj. $R^2$ of multiple linear regressions | | |
|---|---|---|---|
| | $PM_{2.5BL}$ | $SO_{2BL}$ | $NO_{2BL}$ |
| JZ | 0.6776 | 0.4166 | 0.8358 |

| | | | |
|---|---|---|---|
| XN | 0.6899 | 0.0630 | 0.7408 |
| XY | 0.7971 | 0.6741 | 0.8181 |
| JM | 0.7872 | 0.3612 | 0.6480 |
| YC | 0.7168 | 0.2980 | 0.6304 |
| SZ | 0.7175 | 0.3612 | 0.8669 |
| WH | 0.7289 | 0.2718 | 0.6653 |
| EZ | 0.7162 | 0.4592 | 0.7523 |
| HG | 0.6937 | 0.1901 | 0.7220 |
| HS | 0.5695 | 0.2787 | 0.6952 |
| CS | 0.7307 | 0.1255 | 0.7012 |
| YY | 0.7501 | 0.1047 | 0.7592 |
| XG | 0.6755 | 0.4389 | 0.7692 |
| CD | 0.7017 | 0.1730 | 0.6937 |

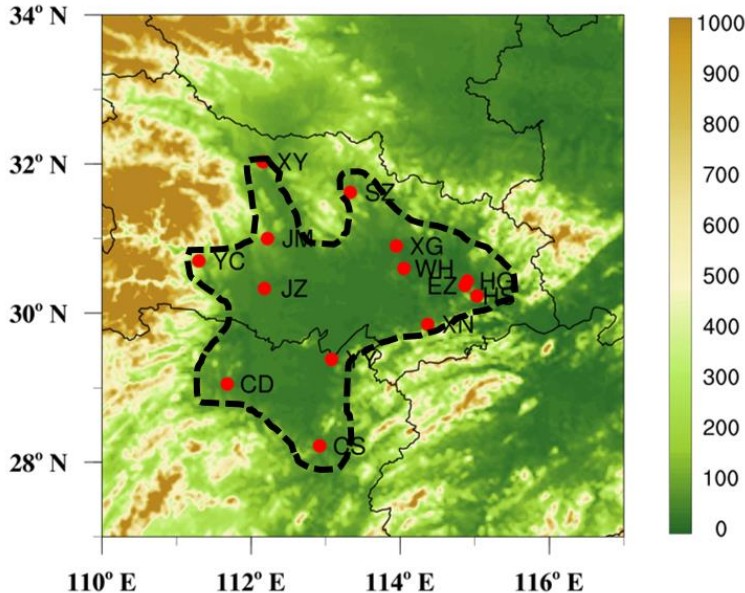

**Figure 1** Topographical height (color contours, m, in a. s. l.) over the THB (outlined with black dashed line) with the locations

of 14 sites (red dots) and the surrounding regions in central China.

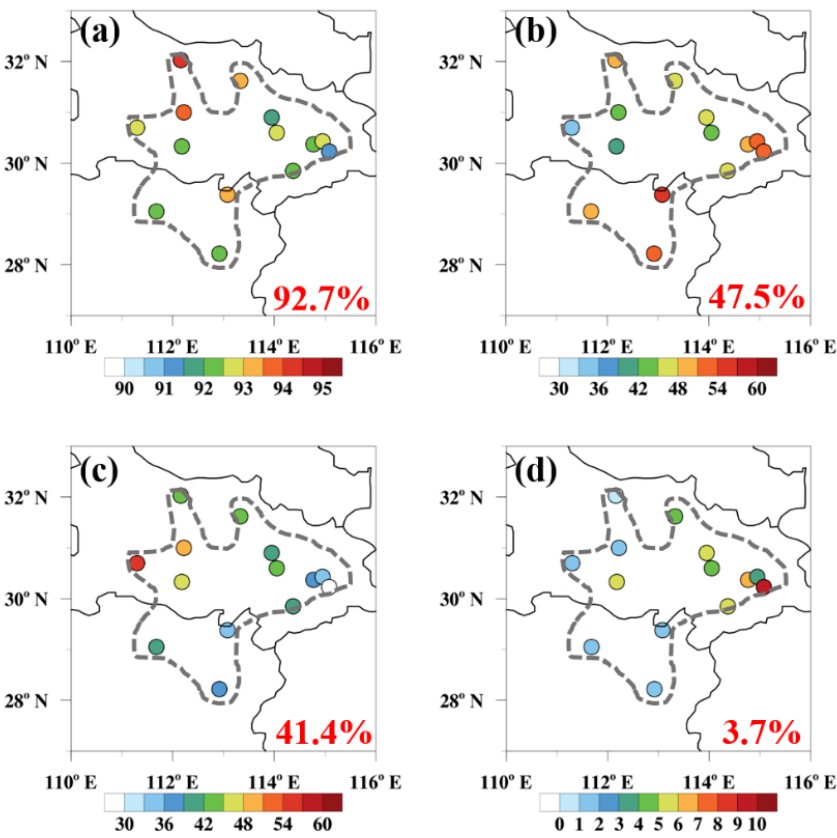

**Figure 2** Spatial distributions of the (a) total and relative contributions of (b) short-term, (c) seasonal and (d) long-term

components to the total variances of daily PM$_{2.5}$ changes observed at 14 sites in the THB with the regional averages of 92.7%,

47.5%, 41.4% and 3.7%.

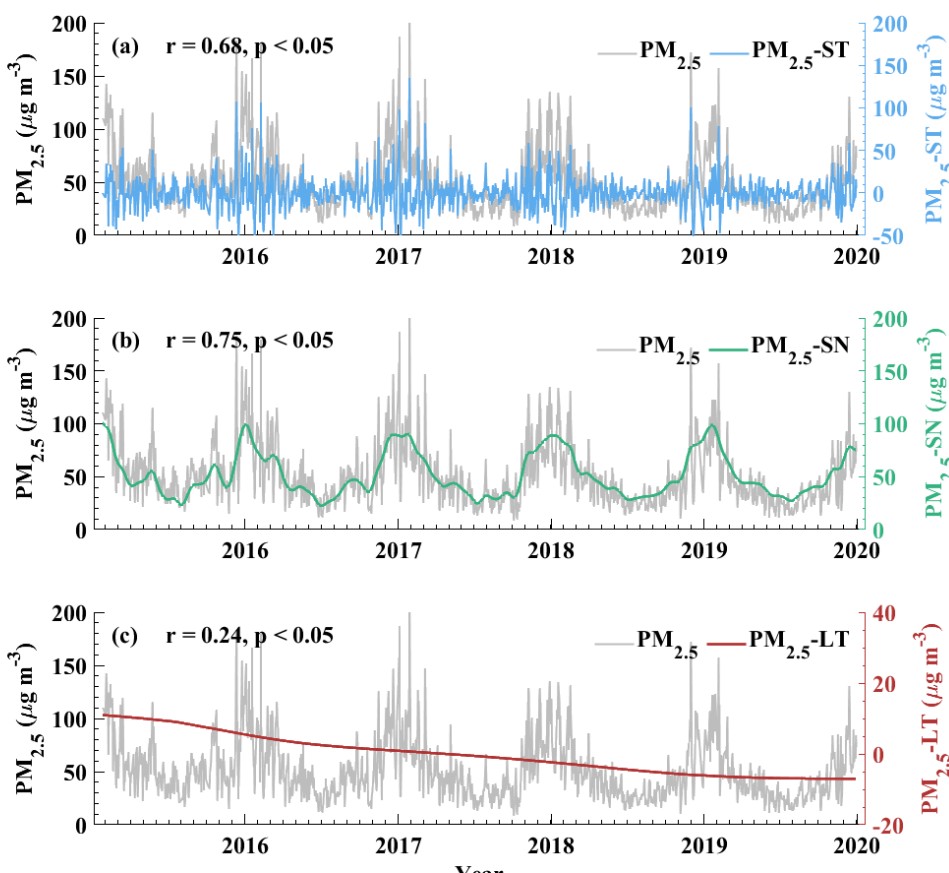

**Figure 3** The relations of regional averages of (a) short-term (PM$_{2.5}$-ST), (b) seasonal (PM$_{2.5}$-SN) and (c) long-term (PM$_{2.5}$-LT) components with the observed daily PM$_{2.5}$ concentrations (PM$_{2.5}$) over the THB from 2015 to 2019.

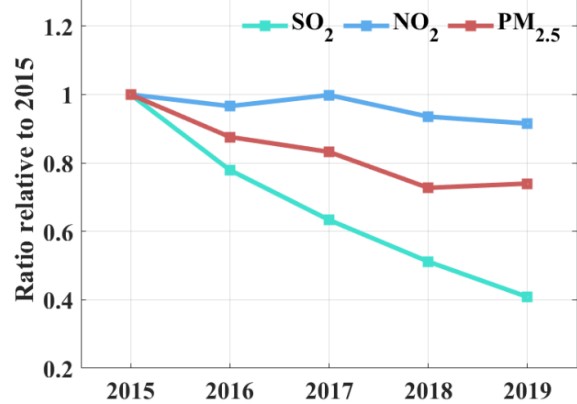

**Figure 4** Interannual variations in the ratios of observed annual mean concentrations of SO$_2$, NO$_2$ and PM$_{2.5}$ relative to those in 2015 averaged over the THB.

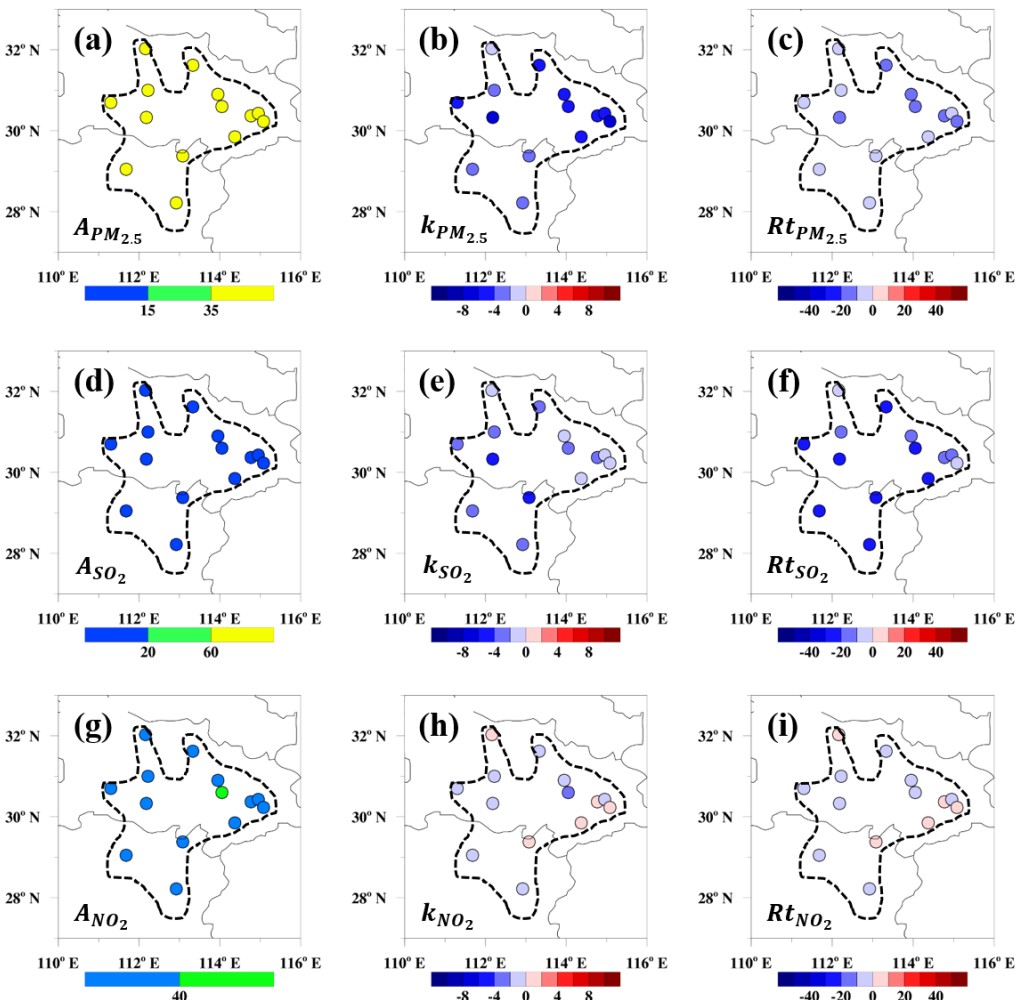

**Figure 5** Spatial distributions of (left column) 5-year averages of (a) $PM_{2.5}$, (d) $SO_2$ and (g) $NO_2$ concentrations ($A$, unit: $\mu g\ m^{-3}$), (middle column) the linear trends in interannual variations of (b) $PM_{2.5}$, (e) $SO_2$ and (h) $NO_2$ ($k$, unit: $\mu g\ m^{-3}\ yr^{-1}$), as well as (right column) the change rates ($Rt=k/A$, unit: % $yr^{-1}$) of (c) $PM_{2.5}$, (f) $SO_2$ and (i) $NO_2$ in the THB over 2015–2019.

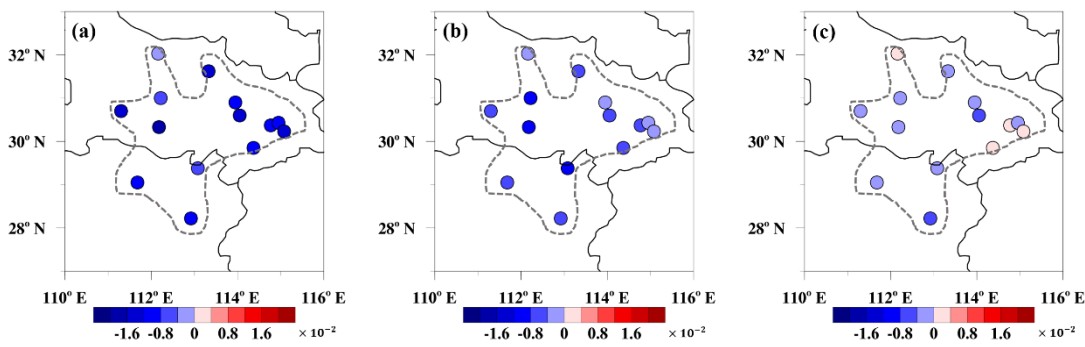

**Figure 6** Spatial distributions of the linear trends in emission-related long-term components of (a) $PM_{2.5}$, (b) $SO_2$ and (c) $NO_2$ (unit: $\mu g\ m^{-3}\ d^{-1}$) over 2015–2019 in the THB

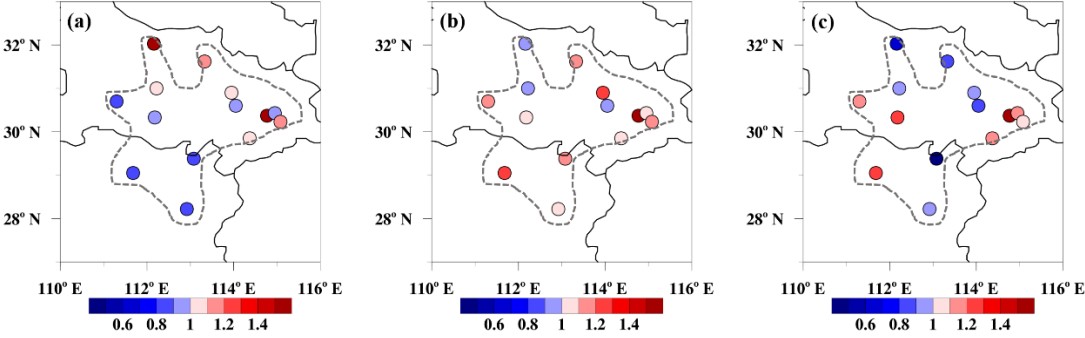

**Figure 7** Spatial distributions of the ratios of linear trends in long-term components ($k_{LT}$) and emission-related long-term

components ($k_{emiss}$) of (a) PM$_{2.5}$, (b) SO$_2$ and (c) NO$_2$ at 14 sites in the THB over 2015–2019.

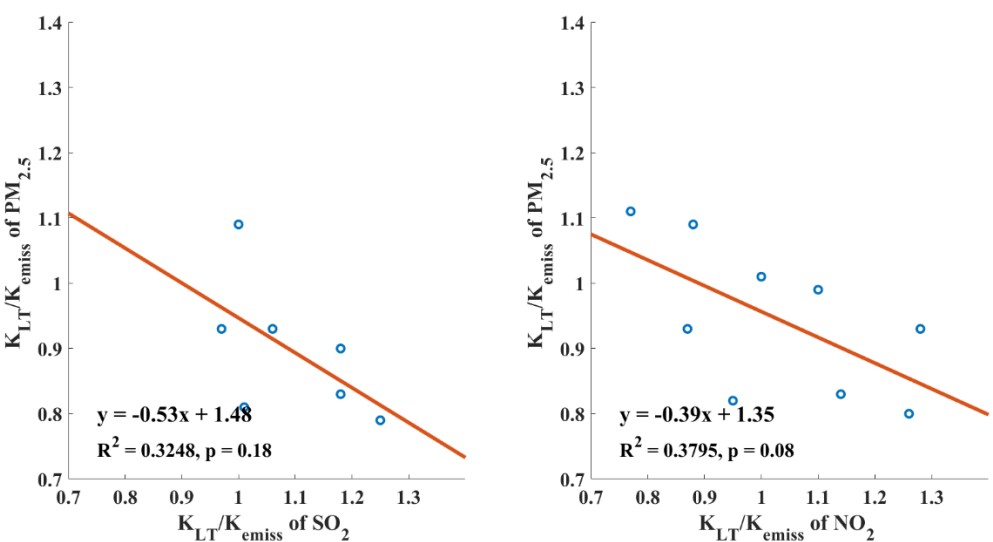

**Figure 8** Scatter plots of the ratios between $k_{LT}$ and $k_{emiss}$ of (a) SO$_2$, (b) NO$_2$ and PM$_{2.5}$ in the THB from 2015 to 2019

with red lines for the linear fitting equations.

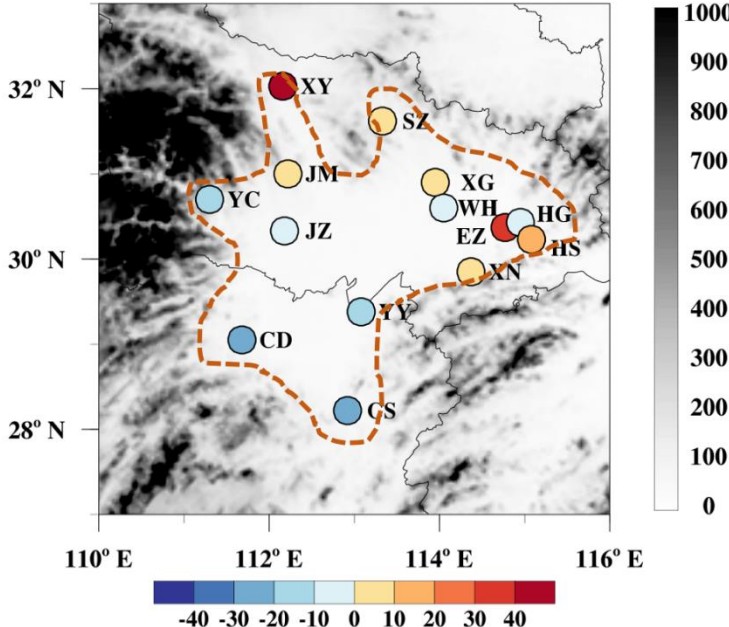

**Figure 9** Spatial distribution of contribution rates (colored dots, unit: %) of meteorological variations to PM$_{2.5}$ reductions with

topographical height (color contours, m, in a. s. l.) in the THB (outlined with orange dashed line) and surrounding regions

from 2015 to 2019.

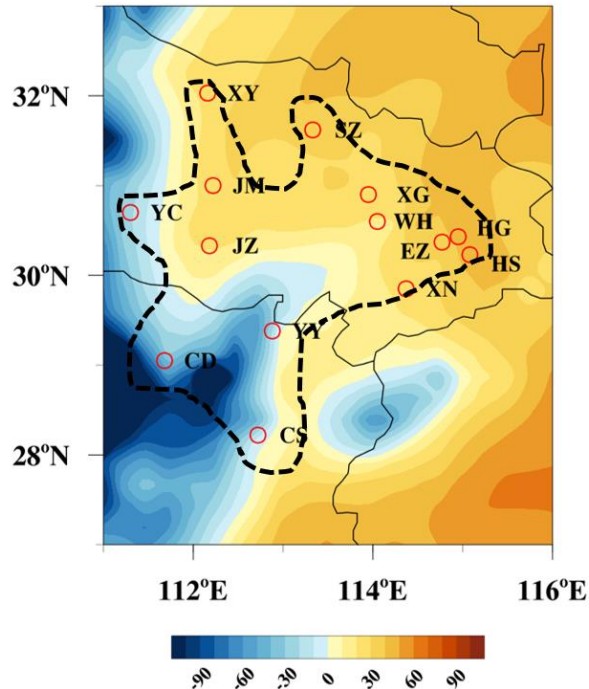

**Figure 10** Spatial distribution of contribution rates of meteorological variations to PM$_{2.5}$ reductions based on WRF-Chem

modeling experiments (contour, unit: %) in the THB outlined with black dashed line and surrounding regions for December

of 2015–2019.