# Peer review of "Meteorological effects on PM2.5 change over a receptor region in regional transport of air pollutants: observational study of recent year emission reduction in central China"

_Atmospheric Chemistry and Physics, 2021_

## Author Comment (AC1)

Dear Editors and Referees:

Thank you very much for your careful review and constructive comments on our manuscript acp-2021-709. We have accordingly made the substantial revisions. The revised portions are highlighted in the revised manuscript. In the following, we quoted each review question in the square brackets and added our response after each paragraph.

**Responses to Referee #1**

[*1. General comments: The manuscript by Sun et al. analyzes the impact of meteorological factors on the changes in $PM_{2.5}$ concentration in Twain-Hu Basin, China using a Kolmogorov-Zurbenko filter. They conclude that interannual and seasonal meteorology have the largest impacts on the changes in $PM_{2.5}$. However, the method used in this work is not validated using synthetic data or with other methods, so the accuracy of the results is doubtful. In addition, this work focuses on a very small region, and the results may not have broader implications for national - global air pollution issues and may not fit the scope of the general ACP readership.]*

**Response 1.1:** Thanks for the referee's comments and suggestions. Please find our response as follows and the subsequent **Response 1.2** to the referee's comments and suggestions:

According to the suggestions, we have conducted the simulation experiments with Weather Research and Forecasting model with Chemistry (WRF-Chem) to validate the accuracy of the results with KZ filter, which is added in the new *Sect. 3.6* as follows:

**3.6 Meteorological contribution to $PM_{2.5}$ changes validated with WRF-Chem modeling**

The above observational study investigated the meteorological influence on the changes in $PM_{2.5}$ concentrations in the THB using KZ filter, with concluding the large impact of meteorology on the $PM_{2.5}$ changes over 2015–2019. To validate this conclusion of analyses with KZ filter, we designed three sets of modeling experiments CTRL, SENS-MET and SENS-EMI (Table S6) for December of 2015–2019, respectively driven with the changing meteorology and anthropogenic emissions over 2015–2019, the fixed meteorological conditions and anthropogenic emissions of 2015 with atmospheric chemical model

WRF-Chem (Weather Research and Forecasting model with Chemistry). Air pollutant emission inventories, modeling configuration, experiment design and modeling verification were described in the supplement. The modeling verification of experiments CTRL indicated that $PM_{2.5}$ and meteorology were reasonably reproduced by the WRF-Chem simulation (Figs.S4–S5, Table S7), and the designed three sets of modeling experiments CTRL, SENS-MET and SENS-EMI could be used in the further analyses of emission and meteorological impact on $PM_{2.5}$ change over 2015–2019 to confirm the results of KZ filter.

We derived the effect of meteorology by comparing the simulated $PM_{2.5}$ concentrations in the three sets of experiments CTRL, SENS-MET and SENS-EMI (Table S6). The relative contribution of meteorology to the interannual changes of $PM_{2.5}$ concentrations was calculated with a linear additive relationship of contributions of meteorology and emission in the following equations:

$$Con_{MET} = \frac{k_{MET}}{k_{CTRL}} \tag{11}$$

$$Con_{EMI} = \frac{k_{EMI}}{k_{CTRL}} \tag{12}$$

$$RCon_{MET} = \frac{Con_{MET}}{Con_{MET} + Con_{EMI}} \times 100\% \tag{13}$$

$k_{CTRL}$, $k_{MET}$ and $k_{EMI}$ represent the trends in interannual changes of $PM_{2.5}$ concentrations simulated by the experiments CTRL, SENS-MET and SENS-EMI, respectively. $Con_{MET}$ and $Con_{EMI}$ are the contribution of meteorology and emission, and $RCon_{MET}$ is the contribution rate (%) of meteorology to interannual changes of $PM_{2.5}$ concentrations (Zhang et al., 2020).

Based on WRF-Chem modeling experiments, we assessed the impact of meteorological changes on interannual $PM_{2.5}$ variations from 2015 to 2019 with *Eqs. (11–13)*. The relative contribution of meteorology to interannual $PM_{2.5}$ variations displayed the regional pattern of northern positive and southern negative values over the THB (Fig. 10), confirming the impact of meteorological changes by accelerating and offsetting the effects of emission reductions on $PM_{2.5}$ declining trends in the northern and southern THB, respectively. The general spatial distribution of meteorological contribution rates to $PM_{2.5}$ declining trends from the WRF-Chem simulation was consistent with the results using KZ filter (Figs. 9 and 10), validating the results with KZ filter that meteorological drivers exerted a contrary impact of northern positive and southern negative contribution on long-term changes of $PM_{2.5}$ concentrations in the THB.

[Figure]

**Figure 10** Spatial distribution of contribution rates of meteorological variations to PM$_{2.5}$ reductions based on WRF-Chem modeling experiments (contour, unit: %) in the THB outlined with black dashed line and surrounding regions for December of 2015–2019.

*[1. General comments: ... ... ... ... In addition, this work focuses on a very small region, and the results may not have broader implications for national - global air pollution issues and may not fit the scope of the general ACP readership]*

**Response 1.2:** In response to the above comments, we have clarified the highlights and implications in the revised *Abstract* and *Introduction* as follows:

The THB covering a large region of two provinces, Hubei and Hunan in central China, is surrounded by the high air pollutant emission regions in North China Plain (NCP) to the north, Yangtze River Delta (YRD) to the east, Pearl River Delta (PRD) to the south and Sichuan Basin (SB) to the west (Lin et al., 2018). Driven by East Asian monsoonal winds over Central Eastern China, THB is a major receptor region in regional transport of air pollutants over China (Shen et al., 2020). Governed by the multi-scale atmospheric circulations, air pollutants emitted from the upwind source regions can be transported easily to the downstream receptor region exacerbating the regional air quality, which can result in a complicated relation of source and receptor in regional transport of air pollutants (Hu et al., 2021). However, the previous studies mostly focused on the atmospheric environment change in the source regions with high anthropogenic emissions of air pollutants, and there have been few assessments on multi-scale changes of atmospheric environment over the receptor region in regional transport of air pollutants. In the present

study of 5-year observations and modeling, we targeted the THB, a large region of heavy $PM_{2.5}$ pollutions over central China, to assess the meteorological effect on $PM_{2.5}$ changes over a receptor region in regional transport of air pollutants, and we assessed the contributions of air pollutant emissions and meteorological conditions to air quality change over this receptor region with the long-term observations over recent years. Our results highlight the effects of emission mitigation and meteorological changes on source-receptor relationship of region transport of air pollutants with the implication of long-range transport of air pollutants for regional and global environment changes. Therefore, the results in this paper have broader implications for regional - global air pollution issues and fit the scope of the general ACP readership.


*[3. L99-100, please clarify what the numbers in the subscript of KZ stand for and why using 1.7 years here.]*

**Response 3:** The KZ filter $KZ_{m,p}$ is a low-pass filter based on an iterative moving average to remove the high frequency variations from the daily observational data, *m* and *p* in the subscript of KZ are moving average (unit: day) and number of iterations (unit: time) respectively.

By comparing different sets of moving average *m* and number of iterations *p*, it was found that the decomposed time series using $KZ_{15,5}$ (15-day length with five iterations) filter exhibited no white noise (short-term component), and the trend of long-term component derived with $KZ_{365,3}$ (365-day length with three iterations) filter corresponded approximately to the interannual trend of the original data, so that $KZ_{15,5}$ and $KZ_{365,3}$ filters were used to decompose the short-term and long-term components from the daily observational data (Rao and Zurbenko, 1994; Eskridge et al., 1997).

Based on the spectral decompositions of the daily observational data and three components (Fig. R1), the power spectral of daily observational data in periods less than 33 days and longer than 632 days (1.7 years) have been well reproduced by short-term and long-term components, and seasonal component represents well the seasonal variations, i.e., periods between 33 days and 1.7 years. We also clarified why using 33 days and 1.7 years in the revised manuscript (Lines 106–113) as shown below:

By comparing different sets of moving average *m* and number of iterations *p*, it was found that the decomposed time series using $KZ_{15,5}$ filter exhibited no white noise (short-term component), and the trend of long-term component derived with $KZ_{365,3}$ filter corresponded approximately to the interannual trend of the original data (Rao and Zurbenko, 1994; Eskridge et al., 1997). Based on the spectral decompositions of the daily observational data and three components, the power spectral of daily observational data in periods less than 33 days and longer than 632 days (1.7 years) have been well reproduced by short-term and long-term components, and seasonal component represents well the seasonal variations, i.e., periods between 33 days and 1.7 years (Seo et al., 2018). Thus we applied $KZ_{15,5}$ and $KZ_{365,3}$ filters to remove the variations with the periods shorter than 33 days and 1.7 years in this study.

[Figure]

**Figure R1** Power spectra of (a) log-transformed original time series X (black line) and (b) the short-term (less than 33 days), (c) seasonal (between 33 days and 632 days), and (d) long-term components (longer than 632 days) (red lines). Effective filter widths for $KZ_{15,5}$ filter (33 days) and $KZ_{365,3}$ filter (632 days) are marked with blue vertical dashed lines. The power spectrum of the original time series in (a) is represented with gray lines in (b-d) (Seo et al., 2018).


[revised manuscript text omitted]

*[6. L196, what are the relative contributions of emissions and meteorology to the long-term changes in PM$_{2.5}$ based on the analyses here?]*

**Response 6:** We applied $KZ_{15,5}$ and $KZ_{365,3}$ filters to remove variabilities of periods shorter than 33 days and 1.7 years and decompose the daily environmental data into short-term, seasonal and long-term components. The long-term component can be further separated into emission-related and meteorology-related components by isolating the emission-related component using a multiple linear regression model with representative meteorological variables (Seo et al., 2018). The detailed methods about the separation of emission- and meteorology-related long-term components are displayed in Fig. R2 and *Sect. 2.3* of the revised manuscript.

The slope of the long-term component can reveal the long-term trend after short-term and seasonal variations are removed from the daily observational data. The difference between the slope of emission-related long-term and long-term components of PM$_{2.5}$ is caused by meteorological changes. The meteorological contribution to the PM$_{2.5}$ declining trend is quantitatively assessed with Eq. (10) in the revised manuscript (Lines 338–340) as follows:

$$\text{Con}_{\text{met}} = \frac{k_{\text{LT}} - k_{\text{emiss}}}{k_{\text{LT}}} \times 100\%. \tag{10}$$

$\text{Con}_{\text{met}}$ (in %) is estimated with the linear trends $k_{\text{LT}}$ of long-term component $PM_{2.5LT}(t)$ and $k_{\text{emiss}}$ of emission-related long-term component $PM_{2.5LT}^{\text{emiss}}(t)$.

[Figure]

**Figure R2** Schematic flowchart of time series decomposition of any environmental variable X into short-term, seasonal, and emission-related and meteorology-related long-term components (Seo et al., 2018).

trend of anthropogenic emissions estimated from emission inventories can support the explanation of the changes in air pollutant concentrations.

[Figure]

**Figure S2** (a) Interannual variations in the ratios of MEIC emissions for 2010–2017 compared with satellite- and ground- based observations relative to those in 2013 (Zheng et al., 2018), (b) interannual variations in the ratios of annual total emission of $SO_2$, $NO_x$ and PM relative to those in 2015 averaged over the THB reported by National Bureau of Statistic of China.

**Reference:**

Zheng, B., Tong, D., Li, M., Liu, F., Hong, C. P., Geng, G. N., Li, H. Y., Li, X., Peng, L. Q., and Qi, J.: Trends in China's anthropogenic emissions since 2010 as the consequence of clean air actions, Atmospheric Chemistry and Physics, 18, 14095-14111, 2018.

---

## Author Comment (AC2)

Dear Editors and Referees:

Thank you very much for your careful review and constructive comments on our manuscript acp-2021-709. We have accordingly made the substantial revisions. The revised portions are highlighted in the revised manuscript. In the following, we quoted each review question in the square brackets and added our response after each paragraph.

**Responses to Referee #2**

[*1. General comments: This study investigates the relative contribution from meteorological effect and emission changes to PM$_{2.5}$ variation over the Twain-Hu Basin (THB) based on the Kolmogorov–Zurbenko (KZ) filtering of long-term air quality measurement data. It is indicated that the reduction in anthropogenic emissions was the primary cause for the long-term decline in PM$_{2.5}$ concentrations and the meteorological changes moderated the PM$_{2.5}$ variations in the THB. However, in terms of novelty and broad interest, this work still needs to be improved. Besides, there could be great uncertainties associated with the multiple linear regression and KZ filtering method, but the authors have not validated the method and touched on the uncertainties in the conclusion. Here list some of my main concerns.*]

**Response 1.1:** Thanks for the referee's comments and suggestions. Please find our response as follows and the subsequent **Response 1.2** to the referee's comments and suggestions.

We have clarified the highlights and implications for novelty and broad interest in the revised *Abstract* and *Introduction* as follows:

The THB covering a large region of two provinces, Hubei and Hunan in central China, is surrounded by the high air pollutant emission regions in North China Plain (NCP) to the north, Yangtze River Delta (YRD) to the east, Pearl River Delta (PRD) to the south and Sichuan Basin (SB) to the west (Lin et al., 2018). Driven by East Asian monsoonal winds over Central Eastern China, THB is a major receptor region in regional transport of air pollutants over China (Shen et al., 2020). Governed by the multi-scale atmospheric circulations, air pollutants emitted from the upwind source regions can be transported easily to the downstream receptor region exacerbating the regional air quality, which can result in a complicated relation of source and receptor in regional transport of air pollutants (Hu et al., 2021). However, the

previous studies mostly focused on the atmospheric environment change in the source regions with high anthropogenic emissions of air pollutants, and there have been few assessments on multi-scale changes of atmospheric environment over the receptor region in regional transport of air pollutants. In the present study of 5-year observations and modeling, we targeted the THB, a large region of heavy $PM_{2.5}$ pollutions over central China, to assess the meteorological effect on $PM_{2.5}$ changes over a receptor region in regional transport of air pollutants, and we assessed the contributions of air pollutant emissions and meteorological conditions to air quality change over this receptor region with the long-term observations over recent years. Our results highlight the effects of emission mitigation and meteorological changes on source-receptor relationship of region transport of air pollutants with the implication of long-range transport of air pollutants for regional and global environment changes. Therefore, the results in this paper have broader implications for regional - global air pollution issues.

*[1. **General comments:** ... ... ... ... Besides, there could be great uncertainties associated with the multiple linear regression and KZ filtering method, but the authors have not validated the method and touched on the uncertainties in the conclusion.]*

**Response 1.2:** The multiple linear regression is done stepwise, adding and deleting meteorological factors based on their independent statistical significance to obtain the best regression fit for air pollutants. For meteorological variables not in the final multiple linear regression model, the regression coefficients are zero. The selected meteorological variables differ by sites and all regression coefficients pass the confidence of 99%. The multiple linear regressions explained $PM_{2.5BL}$, $SO_{2BL}$ and $NO_{2BL}$ with adjusted determination coefficients (Adj. $R^2$) of 0.5695–0.8093, 0.0630–0.4592 and 0.6304–0.8669 passing the confidence level of 99 % in all the THB sites, confirming the reasonable construct of multiple linear regressions. The detailed justification and validation of selecting the meteorological parameters and discussions about validating the multiple linear regressions are clarified in *Sect. 3.2* of the revised manuscript.

To verify the results using KZ filter, we have added more discussions by clarifying the reasonable decomposition of multi-time scale components in Lines 164–167 and Lines 179–184 based on the previous studies as follows:

[revised manuscript text omitted]

*[2. There are many parameters used in KZ filtering and multiple linear regression. The justification and validation of the selection of them should be provided. I think the changes in data coverage or the parameter selection would largely influence the final quantitative estimation of contributions, which is suggested to be elaborated.]*

**Response 2:** Following the reviewer's suggestion, we have justified and validated the selection of

meteorological parameters in *Sect. 3.2* (Lines 215–221 and Lines 234–241) as follows:

Based on our understanding of chemical and physical processes of diffusive transport, chemical transformation, emissions and depositions of $PM_{2.5}$ in the atmosphere, the dominant meteorological factors for changing $PM_{2.5}$ concentrations over china are wind speed, relative humidity, air temperature, atmospheric pressure and precipitation (Chen et al., 2020). We examined the significant correlations between baseline components of air pollutant concentrations and selected a set of meteorological factors, including air temperature, wind speed, precipitation, relative humidity, and air pressure (Tables S1-S3 in the *Supplement*). The meteorological parameters selected in this study are consistent with the previous studies (Chen et al., 2020). (Lines 215–221)

… … … …

The multiple linear regression is done stepwise, by adding and deleting meteorological factors based on their independent statistical significance to obtain the best regression fit for air pollutants (Draper, 1998). The multiple linear regressions explained $PM_{2.5BL}$, $SO_{2BL}$ and $NO_{2BL}$ with adjusted determination coefficients (Adj. $R^2$) of 0.5695–0.8093, 0.0630–0.4592 and 0.6304–0.8669 passing the confidence level of 99 % in all the THB sites, confirming the reasonable construct of multiple linear regressions. (Lines 234–241)

Following the reviewer's comments, we have elaborated that the changes in data coverage or the parameter selection would largely influence the final quantitative estimation of contributions of meteorology and emissions for the limitation and outlook of our study in the revised *Conclusions* (Lines 429–435) as follows:

The changes in data coverage and the meteorological parameter selection would largely influence the final quantitative estimation of contributions of meteorology and emissions. Due to the limitation of the data coverage of observational data, further work could be desired with climate analyses of long-term data of fine meteorological and environmental observations and more comprehensively modeling of chemical and physical processes in the atmosphere to generalize the assessment on the effects of emission mitigation and meteorological changes on source-receptor relationship of region transport of air pollutants.


It seems not very likely that the variation in synoptic weather or meteorological conditions has such a large heterogeneity at such a small spatial scale over EZ, HG and HS. However, the underlying surface conditions dominate the near-surface meteorological conditions in the atmospheric boundary layer at a

small scale (Wang et al., 2017). The topography and land use of HG, HS, EZ and surrounding regions vary distinctly with underlying surface conditions of plain, lakes and hilly area (Fig. R1). The underlying surface of observational sites with different near-surface meteorology effectively influence the local accumulation, chemical transformation, dry and wet depositions of air pollutants (Bai et al., 2022). Therefore, the heterogeneity of meteorological contribution to PM$_{2.5}$ at such a small spatial scale might be attributed to the local meteorological conditions in the atmospheric boundary layer, which is largely affected by the underlying surface changes.

[Figure]

**Figure R1** Distribution of (a) topographical height (color contours, m, in a. s. l.) and (b) land use over HG, EZ, HS and the surrounding regions in the THB (https://lpdaac.usgs.gov/products/mcd12q1v006/, last access: January 17, 2022).

---

## Author Response (AR2)

Dear Editors and Referees:

Thank you very much for your careful review and constructive comments on our manuscript acp-2021-709R. We have accordingly made the revisions. The revised portions are highlighted in the revised manuscript. In the following, we quoted the suggestion for revision in the square brackets and added our response after the suggestion.

**Responses to Referee #3**

[**Comments**: *The authors have addressed most of my concerns with additional analysis and discussions. The majority of the authors' response somehow admitted the uncertainties or even biases introduced by the method and representativeness of the observational sites. For instance, the unresolvable roles of emission variations of primary aerosol, gas precursors, and chemical transformations. I do think that the uncertainties and the validity of their argument ought to be clearly demonstrated. After these minor corrections, I would recommend its publication on Atmospheric Chemistry and Physics.*]

**Response:** Thanks for the referee's encouraging comments and helpful suggestions. We have clarified that the emission reductions of primary aerosols and gaseous precursors dominate the $PM_{2.5}$ decline, and the decreasing emissions of gaseous precursors drove enhancing the chemical transformation of $NO_2$ and $SO_2$ efficiently to secondary $PM_{2.5}$, which might offset the effect of emission reduction on $PM_{2.5}$ decline. To better demonstrate our results, we have added the according discussions in the revised manuscript (lines 318–331 and lines 423–425) as follows:

The decreasing emissions of gaseous precursors drove faster oxidation of $NO_2$ and $SO_2$ to nitrate

and sulfate components of $PM_{2.5}$ in the source regions of air pollution in China (Zhai et al., 2021; Huang et al., 2021). This study identified the enhancing contribution of gaseous precursors to $PM_{2.5}$ concentrations with reducing anthropogenic emissions of air pollutants over the receptor region in regional $PM_{2.5}$ transport.

There are a few sources of uncertainty in the discussion of chemical transformation, for example in the separation of emission- and meteorology-related long-term components and in the selection of observational sites. Our results point to better understand the offsetting effect of $SO_2$ and $NO_2$ oxidized to secondary particles on $PM_{2.5}$ mitigation during the emission reduction in the THB … … … …Further work with long-term observational data of $PM_{2.5}$ components like black and organic carbon could be conducted to quantify the influence of emissions of primary components and chemical transformation of gaseous precursors on $PM_{2.5}$ changes. (lines 318–331)

We took considered the uncertainties induced by statistical methods as systematic biases and explained the offsetting effect of enhancing oxidation of gaseous precursors to secondary particles on $PM_{2.5}$ decline during the stringent emission controls. (lines 423–425)

**References:**

Huang, X., Ding, A., Gao, J., Zheng, B., Zhou, D., Qi, X., Tang, R., Wang, J., Ren, C., and Nie, W.: Enhanced secondary pollution offset reduction of primary emissions during COVID-19 lockdown in China, National Science Review, 8, nwaa137, 2021.

Zhai, S., Jacob, D. J., Wang, X., Liu, Z., Wen, T., Shah, V., Li, K., Moch, J. M., Bates, K. H., and Song, S.: Control of particulate nitrate air pollution in China, Nature Geoscience, 14, 389-395, 2021.